# Small-scale livelihood and cultural fire: Global spatiotemporal characteristics, and gaps in data

Cathy Smith[1,2], Matthew Kasoar[2,3]*, Oliver Perkins[2,4], James D. A. Millington[2,4], Jayalaxshmi Mistry[1,2]

**1** Department of Geography, Royal Holloway, University of London, Egham, United Kingdom, **2** The Leverhulme Centre for Wildfires, Environment and Society, London, United Kingdom, **3** Department of Physics, Imperial College London, London, United Kingdom, **4** Department of Geography, King's College London, London, United Kingdom

◑ These authors contributed equally to this work.
* m.kasoar12@imperial.ac.uk

## Abstract

Human fire use is a key activity and process in many landscapes and ecosystems around the world, varying spatiotemporally depending on social, economic, and ecological factors. Recently, initiatives have begun to synthesise data on global fire use from across multiple disciplines and disparate sources into coherent databases. Here, we draw on information from one of these databases, the Livelihood Fire Database, which collates data on fire use practices worldwide from case studies in the literature. We examine data from 345 case study locations spanning 69 countries regarding return interval, area burned, and seasonality of anthropogenic fires set to meet small-scale rural livelihood objectives and/or for cultural reasons. We distinguish patterns in the spatiotemporal nature of fires associated with different fire-use purposes, such as clearing vegetation for agriculture, maintaining pasture for livestock, promoting certain plant species for gathering, or driving game when hunting. For many fire uses, especially those related to hunting, gathering, human wellbeing, and social signalling, there are very limited quantitative data available, but it is possible to draw qualitative insights from case studies. Case studies demonstrate that environmental and social conditions drive variation in fire use for the same purpose, reiterating that assumptions of uniform drivers of anthropogenic fire may be misleading. Nonetheless where quantitative data are available, we find some correspondence between the spatiotemporal nature of fires and fire-use purpose, suggesting that distinguishing between different fire-use purposes may be useful to understand and to better model their likely timing, size, and frequency relative to climate and other drivers. We recommend examples where the diagnosis of these broad relationships between fire-use purpose and fire properties could enable improved representation of anthropogenic fire in global land surface models, and aid interpretation of remote sensing data. Many of the smaller fires now being revealed in global burned

**Data availability statement:** The analysis in this manuscript makes use of data from the Livelihood Fire Database (LIFE). The full LIFE database and accompanying metadata are available on Figshare at https://doi.org/10.17637/rh.c.5469993.

**Funding:** The authors are grateful for funding from the Leverhulme Trust under grant RC-2018-023.

**Competing interests:** The authors have declared that no competing interests exist.

area data by new fine-scale remote sensing products are likely human-set; continued collection, collation, and analyses of case study data on human fire use globally will be essential to help interpret this improved detection of small fires, and to ensure appropriate representation of the underlying drivers of human activity when modelling fire regimes.

## 1 Introduction

Worldwide, people shape the seasonality, frequency, extent, and intensity of landscape fires by starting fires and reshaping fuel landscapes [1,2]. Since the late 1990s, anthropologists and human geographers, particularly, have published many case studies describing instances of human fire use across continents (e.g., [3–5]). These demonstrate that controlled fire use retains an important cultural role, and/or an economic role within 'small-scale livelihoods' involving agriculture, pastoralism, forestry, hunting and/or gathering [6–8]. By small-scale livelihoods we mean those relying predominantly on family labour or labour exchange with other households, rather than employment of labourers [9]. The purposes of controlled fires in this context are varied and numerous (e.g., clearing weeds or crop residues to facilitate new crops, enhancing forage for livestock) and often have multiple direct and indirect benefits [3]. Such fires are our focus in this article, and we refer them hereafter as 'small-scale livelihood and/or cultural fires'. Case studies are a vital source of information about these kinds of fire use, given that such fire users do not typically keep written records of their fire use, as do many state agencies, non-governmental organizations, or businesses that use controlled or prescribed fire.

Recently it has become apparent that global estimates of burned area have likely omitted many anthropogenic fires, including small-scale livelihood and/or cultural fires, as these fires are typically smaller than the minimum size that existing global remote sensing products have been able to detect [10–14]. For example, the Moderate Resolution Imaging Spectroradiometer (MODIS) has a 500m spatial resolution, meaning that the smallest fires it can detect are approximately 21 ha. Yet 60% of human fires in a recent global database of human fire [15] are smaller than this. When Ramo et al. [13] compared burned area data for sub-Saharan Africa from the FireCCISFD11 product measured at 20m resolution with the Sentinel-2 satellite, they found an 80% increase in burned area compared to the 500m resolution product MCD64A1 from MODIS. Subsequently, when 30m and 20m spatial resolution data from Landsat and Sentinel 2 were incorporated into the latest Global Fire Emissions Database product (GFED5, [16]), estimated global burned was 61% higher than the previous GFED4s version and 93% higher than MCD64A1. Thus, many smaller fires, likely set and used by people for livelihood purposes, have previously been hidden from global remote sensing products and are only now being revealed to scientists working at a global scale.

Similarly, and possibly consequently, there is limited representation of anthropogenic fire use in current global fire models. At the global scale, fire is commonly

modelled as a module of a dynamic global vegetation model (DGVM), and these tend to represent human fire use and suppression using globally readily available proxies such as population density or GDP [17], treating humans as a static environmental source of ignitions but rarely as intentional actors shaping the fire regime in different ways and with diverse objectives [18]. As a result, there is no representation of the socio-ecological processes that drive human fire use, there has been poor agreement between models, and only limited success in reproducing satellite-derived burned area estimates [19,20]. This leaves us with an incomplete representation of the relative importance of different drivers of fire regimes (e.g., climate versus human activity), which in turn hinders understanding of the adaptations we may need to make to ensure sustainable landscapes under future climatic and socio-economic conditions.

As detection of small fires improves, interpreting the patterns revealed will require understanding of human fire use, given that many of the newly detected fires will likely be human-set. Where insights about human fire use from local case studies have been synthesized at the global scale this has usually been in the form of qualitative analyses exploring, for instance, reasons for burning [6,21] or elements of fire knowledge across Indigenous societies [22]. Quantitative data regarding, for instance, fire return intervals or burned areas associated with different types of fire use, have only been collated [15] and analysed in one previous study [23]. However, there the authors examined the full range of land system actors (subsistence, commercial, state), leading to inevitable simplification in the treatment of small-scale livelihood and/or cultural fire. Most researchers have used top-down approaches to attempt to quantify the influence of human activity on global fire regimes, correlating gridded datasets such as population density, Gross Domestic Product (GDP), or cropland extent with satellite-derived estimates of active fires or burned area. Such studies suggest human influences on fire seasonality [24], frequency [25,26], and burned area [27–29], but do not directly engage with the social and ecological factors that shape fire use for differing purposes in different contexts. Le Page et al. [30], are an exception, having used a small number of case studies to explore how dominant types of anthropogenic burning in specific locations caused shifts in the seasonality of fire activity (as measured from satellite data), relative to expected fire seasonality based on a vegetation dryness index. Improved understanding of spatiotemporal characteristics of small-scale livelihood and/or cultural fires will enable more complete and well-reasoned representations of anthropogenic fire in large-scale (including global) models of vegetation fire, and improve interpretation of the latest high-resolution remote sensing data that can detect smaller fires than previously. These tools could then provide more useful insights for management and policy to ensure sustainable landscapes in the context of a changing climate.

In this article we draw on the Livelihood Fire Database (LIFE) [31], to ask whether different fire-use purposes can serve as indicators of the seasonal timing, fire return intervals, and burned areas of anthropogenic fires. LIFE collates case study data on fire use practices worldwide, and allows us to aggregate fire use practices that share the same reported purpose, in order to investigate the extent to which common properties or generalizable relationships exist across different locations and contexts. In doing so, we explore the potential for quantitative data from field-based case studies to reveal spatiotemporal patterns in human fire use that might inform the interpretation of remote sensing data and broad-scale fire modelling. We also draw on qualitative insights from the case studies that point towards important processes for understanding and modelling anthropogenic fire use from the landscape to global scale.

## 2 Methods

### 2.1 The livelihood fire database

The Livelihood Fire Database (LIFE) [31], on which we based this meta-analysis, is a repository of information about contemporary landscape fire use practices worldwide that meet cultural objectives and/or objectives within 'small-scale' livelihoods (those relying predominantly on family labour or labour exchange with other households, rather than employment of labourers) involving agriculture, pastoralism, hunting, gathering and/or forestry. The LIFE database describes 1648 fire use practices which were recorded in 587 case study locations spanning 86 countries. The database draws on 593 literature sources written or published since 1995 that report on original empirical research carried out since 1990.

When the database was created, separate fire practices were distinguished based on the criteria by which the author(s) of the source literature distinguished between distinct types of fire use – usually based on the purposes of fires (for more detail see [8]). Fire use practices included in the database were practiced at the time of the original research (92%) or within living memory of the research participants (8%). For each practice, LIFE includes qualitative and/or quantitative information about up to 37 social and biophysical variables (depending on the types of data available in each source). In this study we analysed data about fire use purpose(s), months when burning takes place, fire return interval and burned area. The subset of fire use practices with known purpose(s) for which data was available for at least one of these spatio-temporal variables were described at 345 case study locations spanning 69 countries.

LIFE draws on published books and peer-reviewed papers, as well as unpublished reports, Master's, and PhD theses, including non-English language sources. These were identified through a systematic literature review using the Clarivate Analytics Web of Science Core Collection, the catalogue of the Fire Research Institute Library (http://fireresearchinstitute.org/), and the literature cited in six existing global studies of fire use [6,15,21,22,32,33]. Additional sources were found by 'snowball sampling' from the bibliography of each source. Smith et al. [8] provide full details of the literature search methodology and structure of the LIFE database. We re-ran the literature searches in October 2022 and updated LIFE with six additional recently published sources, which we included in this analysis.

## 2.2 Fire-use purposes

Information about fire-use purposes was initially recorded in LIFE as descriptive free text information. Smith et al. [8] used this information to create a hierarchical fire-use purpose classification including eight higher tier categories (agriculture, pastoralism, hunting and fishing, gathering, charcoal and firewood production, movement, promoting human health and wellbeing, and social signals) each of which is associated with several of 29 lower tier categories which we use again here (Table 1, with detailed descriptions given in S1 Appendix). In this study, we looked for spatial and temporal patterns of fires set for each of the lower tier purpose categories. For analyses in this paper, we use classes and information as described in Smith et al. [8], except where explicitly stated otherwise. We did not analyze spatiotemporal data for fire use practices for which Smith et al. [8] recorded the lower tier purpose as 'unknown' or 'other'. Multiple purpose categories were listed against some fire use practices in LIFE, but only where a source explicitly stated that the same ignitions were associated with multiple purposes (in 17% of cases). In these cases, here we included a fire practice in our analysis for each purpose listed.

## 2.3 Analysis of fire return interval and burned area data

Quantitative measures of burned areas (in hectares) and fire return intervals (in years) associated with a fire use practice were recorded in LIFE as average values (mean or median), minimum values, maximum values, and/or approximate values (where a single value was stated in the source without explanation of how it was derived), depending on the kind of data available. At least one of these values was available for burned area for 165 fire use practices with known purpose(s) recorded at 113 case study locations spanning 40 countries, and for fire return interval for 298 fire use practices with known purpose(s) recorded at 210 case study locations spanning 49 countries. The geographical distribution of these case study locations are shown in Fig 1 a, b. Where minimum and maximum values were available, we calculated the range. Very few sources provided average values for burned area or fire return interval. For all but three of our fire-use purpose categories (vegetation clearance fires for swidden or permanent agriculture, and crop residue burning), there were five or fewer fire use practices with mean or median burned area data. For all but one of our fire-use purpose categories (vegetation clearance fires for swidden agriculture), there were five or fewer fire use practices with mean or median fire return interval data. Given these limitations, for each fire use practice for which quantitative data were available, we selected the best available estimate of central tendency to allow for comparison between practices. Where a mean or

**Table 1. Fire-use purpose categories.** The table indicates whether a purpose is the primary reason for burning, or a co-benefit. It also indicates the number of examples of each fire-use purpose (*N*) in the LIFE database. Smith et al. ( [8], Table 1) provide examples from the literature for each lower-tier fire-use purpose category. See S1 Appendix for a more detailed description of each purpose, as well as a description of the geographical distribution of case studies in LIFE.

| Fire use purpose | | Reason | N |
|---|---|---|---|
| *Higher-tier* | *Lower-tier* | | |
| Agriculture (A) | A1. Clear vegetation for swidden or semi-permanent agriculture | Primary purpose | 325 |
| | A2. Clear vegetation for permanent agriculture | Primary purpose | 46 |
| | A3. Clear weeds and/or crop residues during the growing season | Primary purpose | 22 |
| | A4. Clear weeds and/or crop residues after harvest to enable planting | Primary purpose | 133 |
| | A5. Reduce crop pests | Primary purpose or co-benefit | 34 |
| Pastoralism (P) | P1. Clear vegetation to establish new pasture areas | Primary purpose | 18 |
| | P2. Enhance forage for grazing livestock | Primary purpose | 219 |
| | P3. Herd livestock | Primary purpose or co-benefit | 17 |
| | P4. Reduce livestock pests and predators | Co-benefit | 50 |
| Hunting and fishing (HF) | HF1. Create or improve habitat for hunted or fished species | Primary purpose or co-benefit | 24 |
| | HF2. Renew forage to draw hunted or fished species into a particular area | Primary purpose or co-benefit | 67 |
| | HF3. Improve visibility or access specifically for hunting or fishing | Primary purpose or co-benefit | 69 |
| | HF4. Drive animals when hunting | Primary purpose | 65 |
| | HF5. Kill, injure, or tire animals when hunting | Primary purpose | 12 |
| Gathering (G) | G1. Enhance productivity of foraged resources | Primary purpose or co-benefit | 134 |
| | G2. Ease the collection of a foraged resource by improving visibility or access | Primary purpose or co-benefit | 42 |
| | G3. Drive wild bees away from hives for honey collection | Primary purpose | 49 |
| Charcoal and fuelwood production (C) | C1. Produce charcoal | Primary purpose | 17 |
| | C2. Produce fuelwood for gathering, or enable gathering of fuelwood | Primary purpose or co-benefit | 20 |
| Movement (M) | M1. Maintain and open trails and waterways for general access | Primary purpose or co-benefit | 51 |
| Human health & wellbeing (HW) | HW1. Reduce animals that are dangerous to or unwanted by humans | Co-benefit | 64 |
| | HW2. Reduce fuel loads to reduce risk of wildfires at a landscape scale | Primary purpose, or co-benefit | 88 |
| | HW3. Create firebreak using fire to protect, e.g., resources, farms, sacred sites | Primary purpose, or co-benefit | 47 |
| | HW4. Suppress a wildfire (using backing fire to fight fire with fire) | Primary purpose | 6 |
| | HW5. Produce a more aesthetically pleasing landscape, or for enjoyment | Co-benefit | 24 |
| Social signals (S) | S1. Communicate about current activity. | Primary purpose | 17 |
| | S2. Show disapproval or protest (arson). | Primary purpose | 53 |
| | S3. For ritual or ceremonies. | Primary purpose | 16 |
| | S4. Assert or maintain cultural identity. | Co-benefit | 113 |

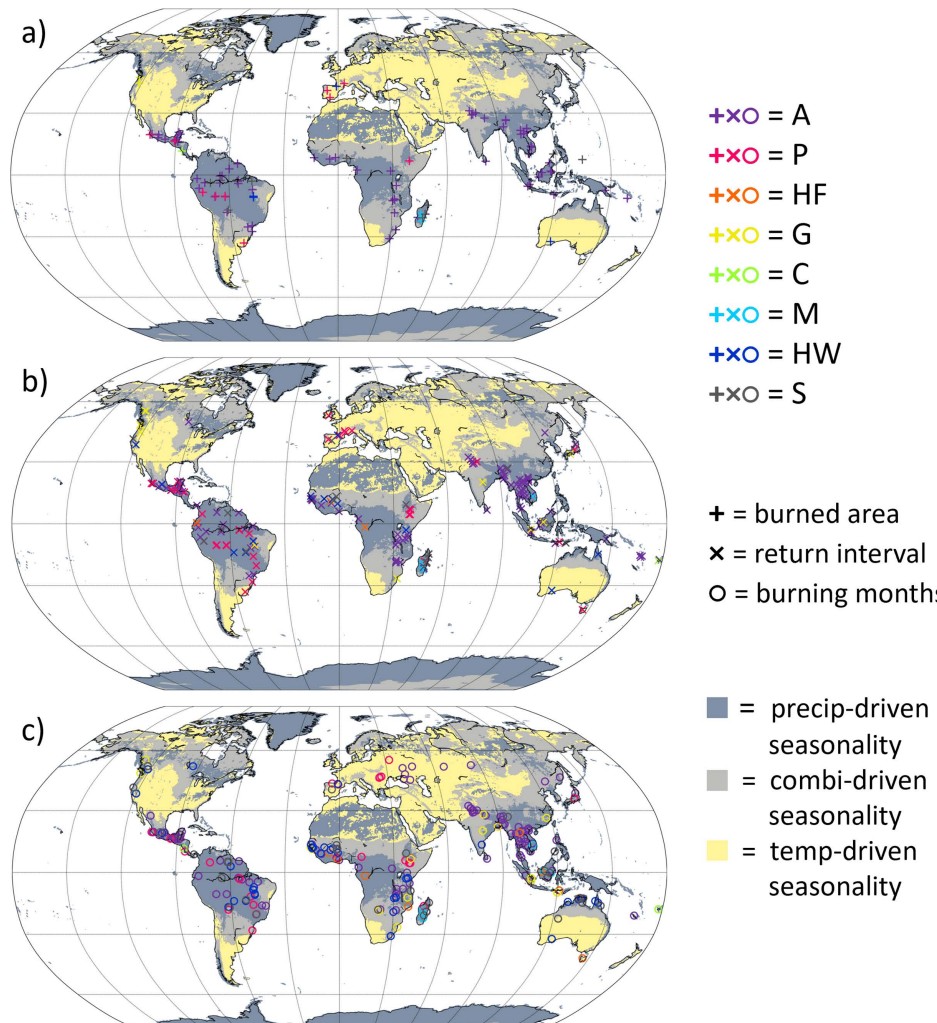

**Fig 1. Maps showing case study locations of fire use practices with known purpose(s) for which spatiotemporal data was available, separated by the type of spatiotemporal data available at each case study location:** a) case study locations with burned area data (165 fire use practices); b) case study locations with return interval data (298 fire use practices); c) case study locations with data on months when burning takes place (572 fire use practices). Background shading indicates seasonality type (precipitation-driven, combination-driven, temperature driven), as described in Section 2.4. The colour of points indicates higher-tier fire use purpose (A = agriculture, P = pastoralism, HF = hunting and fishing, G = gathering, C = charcoal and fuel-wood production, M = movement, HW = health and wellbeing, S = social signals). Where a case study described multiple fire use purposes at the same location, fire use purposes were plotted in the order listed above and so only the last fire use purpose is visible. S1 Fig and S2 Fig in S2 Appendix show maps enlarged over Africa and south-east Asia, to better display these regions where the density of case studies is very high.

median value was available, we selected this, with the mean taking priority over the median where both were available. Failing that, we used the midpoint of the range, and failing that, the approximate value.

## 2.4 Analysis of seasonality data

A process-flow of methods used for seasonality analysis is given in Fig 2. Seasonality information was recorded in LIFE either in the form of months of burning (data available for 572 fire use practices with known purpose(s), spanning 233 case study locations across 57 countries), and/or qualitative descriptions of seasons of burning (data available for 715

fire use practices). To avoid possible subjectivity in the interpretation of qualitative descriptions of the season, for our seasonality analysis here we only make use of the 572 fire use practices for which information was recorded in the form of months of burning. The geographical distribution of the 233 case study locations of these fire use practices for which months of burning were available is shown in Fig 1c.

To perform our seasonality analysis, we first distinguished between three different climate zones: regions where the seasonal cycle is dominated by variation in precipitation, regions where the seasonal cycle is dominated by variation in temperature, and regions where the seasonal cycle involves a combination of both precipitation and temperature varying (see distribution of the three climate zones in Fig 1). We followed the definition of Feddema [34] for this 'seasonality cause' metric, based on the ratio of the annual range of monthly-mean precipitation (P) and potential evapotranspiration (PET):

$$\frac{(P_{max} - P_{min})}{(PET_{max} - PET_{min})} < 0.5 \Rightarrow \textit{Temperature driven seasonality} \tag{1}$$

$$0.5 \leq \frac{(P_{max} - P_{min})}{(PET_{max} - PET_{min})} < 2.0 \Rightarrow \textit{Combination driven seasonality} \tag{2}$$

$$\frac{(P_{max} - P_{min})}{(PET_{max} - PET_{min})} \geq 2.0 \Rightarrow \textit{Precipitation driven seasonality} \tag{3}$$

Note that this metric determines which meteorological quantity (precipitation, temperature, or both) is responsible for the presence of a seasonal cycle, i.e., the main driver of variation over the course of a year. For instance, most tropical areas tend to have precipitation-driven climate seasonality – not because the temperature is low in these regions, but because it

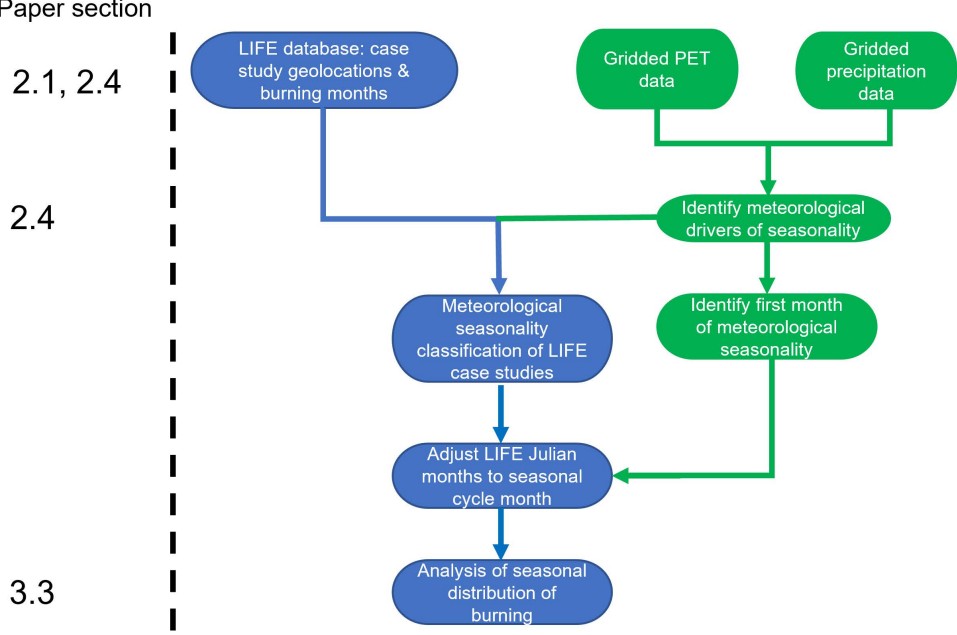

**Fig 2. Process-flow of methods used for seasonality analysis. Potential evapotranspiration (PET) and precipitation data were sourced from ERA5 reanalysis products [35].**

is relatively constant over the year, whereas there is substantial variation in precipitation between the wet and dry seasons. Subtropical and midlatitude areas tend to have temperature-driven climate seasonality, while monsoonal regions as well as some temperate regions are combination-driven. Given the limited number of case study locations in some regions, we used these three seasonality causes as the simplest set of climate zones to aggregate case study locations into, while still capturing the distinct meteorological patterns that may drive different timings in the seasonality of fire use.

We created a 31–year (1990–2020) monthly climatology of precipitation and potential evapotranspiration (PET) from the ERA5 reanalysis [35], which provides these variables globally at a 0.25° × 0.25° spatial resolution. From this climatology, we computed the seasonality cause metric for each land grid cell in the world, and determined which grid cell each case study location falls within, to aggregate case study locations by the local seasonality cause. There is a known issue with the ERA5 diagnostic for PET, whereby in grid cells with either zero, or very dense vegetation cover (deserts, as well as some forest areas with very dense canopy cover) PET is incorrectly diagnosed as zero, meaning that the seasonality cause metric may be incorrectly calculated in these locations. 40 of the fire use practices for which we analysed data, crossing 19 case study locations, potentially fall into such areas (determined from the annual accumulated PE in ERA5 being less than 300 mm, which following Feddema [34] should only occur in exceptionally cold climates). For these 19 case study locations we therefore inspected by eye the precipitation and surface temperature (rather than PET) seasonal cycles in the ERA5 climatology and reclassified the seasonality cause manually based on comparison with similar grid cells.

Having determined the local seasonality cause for each case study location, we analysed the timing of fire use practices relative to the local climatological seasonal cycle, rather than using calendar months. For most fire use purposes, many factors will invariably influence the choice of precisely when to burn, such as weather events, plant and animal behaviour and other environmental indicators, but also past practice and expectations around the timing of seasons. In defining a climatological seasonal cycle for each case study location, we have assumed that insofar as a relationship may exist between the timing of fires associated with a particular fire use purpose and the seasonal cycle, the local climatological seasonal cycle captures enough information to be useful as an indicator of the timing of fire use. The choice of 1990–2020 to define this climatology aligns with the time range of the case studies included in the LIFE database. Although some shifts in regional weather patterns will inevitably have occurred over this time, any shorter time period would be inadequate to robustly characterise the climatological seasons. Additionally, the time of burning is only recorded in LIFE at the calendar month level, and so any variation in timing of burning due to specific weather events is not captured in the underlying data. Our analysis therefore only seeks to define the seasonality of burning at, at most, a monthly time resolution, and we expect that in general, typical seasonal cycles will not have robustly shifted over this period.

409 fire use practices with known purpose(s) fall in precipitation-driven seasonality grid cells. For each of these we defined the local seasonal cycle based on the driest season of the year. At each location we computed a 3-month running mean of the monthly precipitation climatology (with first and last months looped so they were treated as consecutive) and took the centre of the driest 3-month period as the start month of the local seasonal cycle. We translated the calendar months of burning recorded in LIFE into numerical months (−6 to +5) relative to the local start month (= month 0). See S3 Fig in S2 Appendix for graph showing the annual precipitation cycle for each case study location in areas with precipitation-driven seasonality.

Only 23 fire use practices with known purpose(s) fall in temperature-driven seasonality grid cells. For each of these we defined the local seasonal cycle based on the coldest season of the year. We computed a 3-month running mean of PET and took the lowest point as the start month of the local seasonal cycle (month 0). See S4 Fig in S2 Appendix for graph showing the annual PE cycle for each case study location in areas with temperature-driven seasonality.

140 fire use practices with known purpose(s) fall in combination-driven seasonality grid cells. For each of these locations, we calculated the sum of the smoothed precipitation and PET and defined the start month of the local seasonal cycle as being the month that minimized this sum. This corresponds to the overall driest and coldest period, rather than

the individual minimum in either temperature or precipitation (which tends to lag the temperature minimum by 2–3 months in these locations). We again translated the calendar months in LIFE into numerical months relative to this start point (month 0). See S5 Fig in S2 Appendix for graph showing the annual PET and precipitation cycle for each case study location in areas with combination-driven seasonality.

Thus, for each of the three climate seasonality zones we could aggregate all the relevant fire use practices and align the monthly timings of fire use relative to the typical meteorological seasonal cycle rather than the calendar year. This means, for example, that we can compare fire use practices in opposite hemispheres. Relating fire use to the local seasonal cycle allows for more generalized relationships to be identified, which in turn could be used to inform fire modelling or interpret observed trends. The alternative of assuming fixed timings of fire use based on calendar months would require bespoke profiles for different parts of the world, and more importantly does not allow for the possibility that climate and local seasonal cycles may change in future, which could necessitate a change in the timing of fire use. Indeed, some case studies describe changes in the seasonality of burning already occurring, related to anthropogenic climate change (e.g., [36,37]).

## 3 Results

This section presents the findings of the analysis exploring whether and how fire return intervals, burned area and seasonality of fires vary for different fire use purpose categories. We supplement the data with qualitative observations from sources in the LIFE database where they may explain the quantitative results or suggest how our spatiotemporal variables may vary for fire purpose categories for which quantitative data are lacking. For more detail about each of the fire use purpose categories, see S1 Appendix.

The lower tier fire use purpose categories from Smith et al. [8] include both primary, livelihood-oriented, reasons for fire use, such as driving game for hunting or clearing vegetation for swidden agriculture, as well as reasons that are usually co-benefits, such as reducing livestock pests, producing an aesthetically pleasing landscape, or maintaining cultural identity. Primary reasons are more likely to sustain fire use and influence the spatiotemporal patterning of fire [38], and so hereafter we do not present spatiotemporal data for those reasons we deem to usually be co-benefits only (as indicated in Table 1).

Table 2 summarises the number of fire use practices contributing spatiotemporal data for each fire use purpose and within each climate seasonality zone. For each fire-use purpose category, we calculated a mean, median, and standard deviation of the estimates of central tendency for burned area and fire return interval (Table 2). For each fire-use purpose category, we also calculated the mean, median, and standard deviation of the ranges of values for individual fire use practices (Table 2). We only report these statistics and further analyse those fire use purpose categories where there are at least 6 or more fire use practices for which we have a central estimate of burned area and/or fire return interval, or information on months of burning.

### 3.1 Fire return interval

There are limited fire return interval data in the LIFE database: only 12 of our fire use purpose categories have 6 or more fire use practices with fire return interval data (Table 2, Fig 3).

Different types of agricultural fire commonly have different fire return intervals (Table 2, Fig 3). Post-harvest burning of weeds and/or crop residues (A4) is associated with the shortest fire return intervals recorded in LIFE, with the median of the best available estimates of central tendency being 1 year (i.e., annually). Best available estimates of central tendency of fire return intervals for vegetation clearance in swidden or semi-permanent agriculture (A1) range from less than 5 years to more than 30 years, with a median of 8.8 years; longer and wider-ranging fire return intervals than typical for fires of other purpose types. Yet, many studies in LIFE report decreasing fallow periods in swidden agriculture, corresponding to shortening fire return intervals (e.g., [39]). This matches the findings of a previous review of trends in swidden agriculture [40]. While long fallow periods are usually needed to maintain soil fertility, shortening fallows are driven by decreasing

**Table 2. Overview of data available for area burned, fire return interval, and seasonality (i.e., months of burning) for fires set for different purposes.** '*n*' indicates the number of fire use practices with data contributing to the analysis. Mean, median, and standard deviation (SD) are only provided where *n* ≥ 6. For fire use practices with information on months of burning, we separately consider practices located within precipitation-driven, temperature-driven, or combination-driven climate seasonality zones. 'Not applicable' refers to those purposes that are typically co-benefits rather than the primary reason for fire use and would therefore not be expected to directly structure when and where fire is applied to the landscape, or for fire return interval where fires are typically one-off occurrences. See Table 1 for fire-use purpose codes.

| Fire-use purpose | Area burned per ignition (ha) | | | | | | | | Fire return interval (years) | | | | | | | | Seasonality | | |
|---|---|---|---|---|---|---|---|---|---|---|---|---|---|---|---|---|---|---|---|
| | Best available estimate of central tendency | | | | Range of values | | | | Best available estimate of central tendency | | | | Range of values | | | | precip | temp | comb |
| | *n* | Mean | Median | SD | *n* | Mean | Median | SD | *n* | Mean | Median | SD | *n* | Mean | Median | SD | *n* | *n* | *n* |
| A1 | 99 | 1.4 | 1 | 1.3 | 36 | 2.1 | 1.6 | 2.3 | 136 | 10.7 | 8.8 | 6.8 | 81 | 11.1 | 6 | 11.4 | 119 | 0 | 25 |
| A2 | 12 | 1.6 | 1 | 1.4 | 5 | | | | | | | | | | | | 12 | 0 | 5 |
| A3 | 2 | | | | 1 | | | | 4 | | | | 0 | | | | 8 | 0 | 1 |
| A4 | 20 | 1 | 1 | 0.7 | 8 | 2.9 | 1.5 | 3.5 | 45 | 1.7 | 1 | 2.7 | 6 | 3.8 | 3 | 2.7 | 40 | 8 | 21 |
| A5 | 5 | | | | 5 | | | | 5 | | | | 2 | | | | 12 | 0 | 0 |
| P1 | 4 | | | | 0 | | | | | | | | | | | | 3 | 0 | 0 |
| P2 | 22 | 91 | 74.8 | 205.8 | 16 | 102 | 140 | 63.1 | 56 | 3 | 2.5 | 2.2 | 30 | 3.8 | 2.5 | 5.5 | 54 | 6 | 24 |
| P3 | 3 | | | | 3 | | | | 6 | 2.8 | 3 | 1.1 | 6 | 2.8 | 2 | 2.3 | 6 | 0 | 1 |
| P4 | | | | | | | | | | | | | | | | | | | |
| HF1 | 1 | | | | 1 | | | | 3 | | | | 1 | | | | 8 | 0 | 3 |
| HF2 | 1 | | | | 0 | | | | 6 | 2.4 | 1.6 | 2.2 | 3 | | | | 14 | 4 | 8 |
| HF3 | 1 | | | | 0 | | | | 7 | 1.9 | 1 | 1.9 | 2 | | | | 25 | 2 | 6 |
| HF4 | 1 | | | | 1 | | | | 9 | 2.3 | 2.5 | 0.9 | 6 | 2.2 | 2 | 1.3 | 20 | 2 | 3 |
| HF5 | 0 | | | | 0 | | | | 3 | | | | 3 | | | | 5 | 0 | 1 |
| G1 | 6 | 53.8 | 80 | 40.7 | 5 | | | | 23 | 2.5 | 2 | 1.4 | 7 | 2.1 | 2 | 0.7 | 40 | 1 | 14 |
| G2 | 2 | | | | 0 | | | | 6 | 3.2 | 2.5 | 2.9 | 3 | | | | 13 | 0 | 5 |
| G3 | 0 | | | | 0 | | | | 3 | | | | 0 | | | | 11 | 0 | 4 |
| C1 | 0 | | | | 0 | | | | 0 | | | | 0 | | | | 3 | 0 | 1 |
| C2 | 1 | | | | 1 | | | | 1 | | | | 0 | | | | 5 | 0 | 1 |
| M1 | 4 | | | | 3 | | | | 12 | 2.7 | 1.9 | 2.1 | 9 | 2.4 | 2 | 2.1 | 18 | 4 | 3 |
| HW1 | | | | | | | | | | | | | | | | | | | |
| HW2 | 7 | 48.8 | 80 | 39.1 | 5 | | | | 15 | 3.1 | 2 | 2.2 | 9 | 3.7 | 2 | 3 | 30 | 5 | 15 |
| HW3 | 0 | | | | 0 | | | | 6 | 1.9 | 1 | 1.8 | 1 | | | | 19 | 0 | 8 |
| HW4 | 0 | | | | 0 | | | | 0 | | | | 0 | | | | 0 | 0 | 0 |
| HW5 | | | | | | | | | | | | | | | | | | | |
| S1 | 0 | | | | 0 | | | | 0 | | | | 0 | | | | 3 | 0 | 0 |
| S2 | 2 | | | | 0 | | | | 1 | | | | 1 | | | | 4 | 0 | 0 |
| S3 | 0 | | | | 0 | | | | 5 | | | | 0 | | | | 0 | 1 | 4 |
| S4 | | | | | | | | | | | | | | | | | | | |

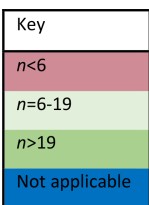

| Key |
|---|
| *n*<6 |
| *n*=6-19 |
| *n*>19 |
| Not applicable |

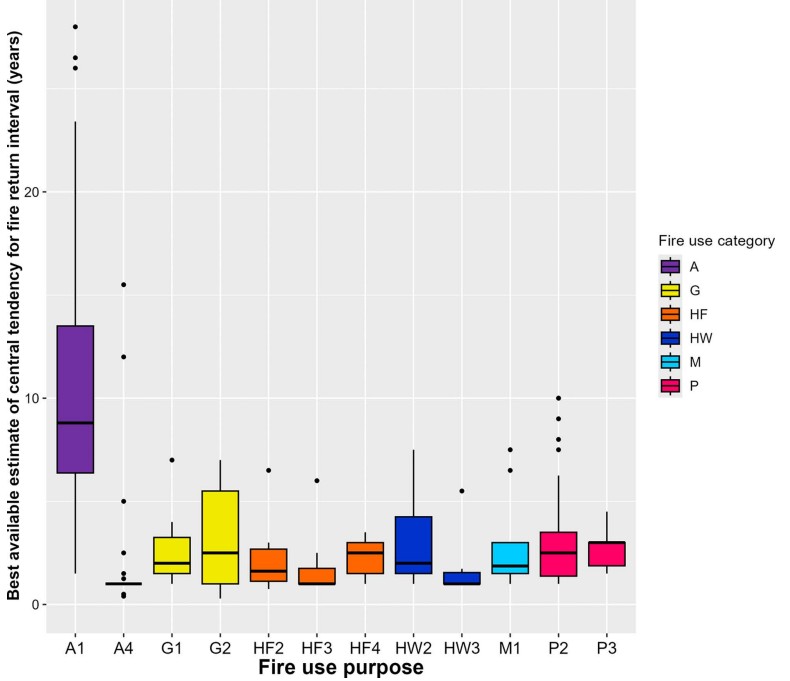

**Fig 3. Boxplots of best available estimates of central tendency for fire return interval by fire purpose category.** The figure only includes boxplots for those fire purpose categories for which there were six or more fire use practices with data on fire return interval. Y-axis limited to exclude outliers.

availability of land for swidden per household, which may relate to population growth, expansion of commercial land uses or protected areas, or diversification of smallholder livelihoods.

For many of the other purpose categories for which we analysed data (P2, P3, G1, G2, HF2, HF3, HF4, HW2, M1) we see broadly similar fire return intervals, with typical means of best available estimates of central tendency being 2–3 years (Fig 3, Table 2). Fire return intervals for fires to create firebreaks (HW3) would seem to be shorter, with typical best available estimates of central tendency being 1–2 years.

Qualitative descriptions in the literature help us to intuit how fire return interval might be expected to vary for some of our other fire use purpose categories. Fires set to establish plots for permanent agriculture (A2) or pasture (P1), or for charcoal production (C1), which follows a line of deforestation in successive years, are typically one-off events. Sometimes we might expect fires to be set annually, for example those associated with honey harvesting (G3), where specific known hives are visited each year (e.g., [41]), or those associated with annual rituals or ceremonies (S3), like the *Calbote* in Spain [42]. For other fire use purposes, such as communication (S1) or arson (S2) we would expect fires to be set on an *ad hoc* basis, rather than at regular intervals.

### 3.2  Burned area

There are very limited burned area data in the LIFE database: only 6 of our fire use purpose categories have 6 or more fire use practices with burned area data (Table 2, Fig 4).

Agricultural fires, for those purposes for which we have data (A1, A2, A4), have consistently smaller burned areas (means of best available estimates of central tendency from 1 to 1.4 ha) and smaller ranges of burned areas within case studies (mean ranges from 2.1 to 2.9 ha) compared with other fire-use purpose types in our classification system (Table 2, Fig 4). This is because they are generally contained within smallholder agricultural plots, which are typically less than

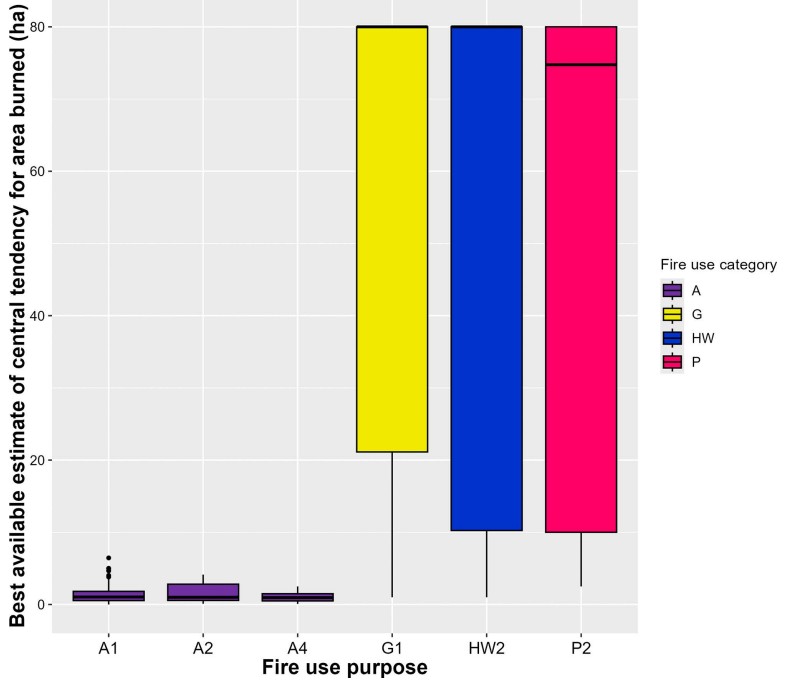

**Fig 4. Boxplots of best available estimates of central tendency for burned area by fire purpose category.** The figure only includes boxplots for those fire purpose categories for which there were six or more fire use practices with data on burned area. Y-axis limited to exclude outliers.

one to two hectares in size. Agricultural burns are usually carefully controlled, for example by using firebreaks or following certain burning patterns relative to the wind, to keep them within the limits of individual plots, especially where there are neighbouring fields close by (e.g., [43]).

By contrast, pasture maintenance fires (P2), fires associated with improving the productivity of foraged plants (G1), and 'patch-mosaic' fires that fragment fuel loads at the landscape scale (HW2) tend to have wider-ranging burned areas, with best available estimates of central tendency ranging from <1 ha to 80 ha or more (Table 2, Fig 4). For each of these fire use purpose categories estimates of central tendency for burned area are positively skewed, with medians of 74.8 ha for P2, 80 ha for G1 and 80 ha for HW2 fire practices across case study locations. These data fit descriptions in the literature suggesting that many fires set for these purposes are broadcast fires which are allowed to burn until they meet natural firebreaks or previously burned areas (E.g., [44]).

Qualitative descriptions in the literature allow us to suggest how we might expect burned area to vary for some of our other fire use purpose categories. For some categories, we would expect great variation in burned area. For example, fires set to improve access to plants for gathering (G2), can be contained fires that are applied to, or around, specific plants such as trees, while others are broadcast fires that are intended to affect vegetation across larger landscape patches (examples of both in Garde [45]). Similarly, across purposes, we would expect hunting and fishing fire use (HF1–5) to be structured by the ecological niche and size of hunted and fished species, resulting in a wide range of possible burned areas. For instance, fire drives of kangaroos in Australia will take place on large scales, likely from tens to hundreds of hectares [46], but fires to drive and kill grasshoppers in Gabon take place on far smaller scales, likely on the order of meters [47]. Fires set to improve general access in the landscape (M1) will also vary significantly in burned area, as they may be carefully contained to clear specific trails or waterways (e.g., [48]) or be broadcast fires set more widely across the landscape, possibly for a variety of more proximate reasons (e.g., [5]). Finally, fires set for ritual and ceremony

(S3), vary from contained fires and bonfires (e.g., [42]), to large broadcast fires like those set at sacred sites in savannas in Burkina Faso and Togo [49].

For other categories, we might expect fires to have generally smaller burned areas. For instance, descriptions of fire use associated with gathering honey (G3), suggest that, unless they are poorly controlled, these usually involve individual branches or bunches of material being burned as a torch to produce smoke, or a contained fire being set around the base of a tree containing a hive (e.g., [50]). Similarly, unless they are poorly controlled, charcoal production fires (C1) will generally be limited to the size of the mounds or pits in which the charcoal is produced (e.g., [51]), and fires set to create firebreaks (HW3) will often be strips of some metres in width, surrounding specific entities such as fruit trees, farm plots or woodlands (e.g., [52]).

### 3.3 Seasonality

We analysed data on the months in which burning takes place for 17 fire use purpose categories in regions with precipitation-driven climate seasonality, for 8 categories in regions with combination-driven climate seasonality, and for only 2 categories in regions with temperature-driven climate seasonality.

As a general pattern, in regions with precipitation or combination-driven seasonality (data are limited for regions with temperature-driven seasonality), fires related to vegetation clearance to establish plots for agriculture (A1 and A2) come later than other forms of fire use, relative to the driest month (Figs 5 and 6). In the three months preceding the driest month, fires for such vegetation clearance purposes are described as taking place during these months for only 6% of fire use practices, compared with 47% of fire use practices associated with other fire use purpose categories. This matches descriptions of agricultural cycles in the literature. Vegetation is typically cut during dry times of year, usually by hand, before being left to dry for several weeks to months. Burning then takes place in the late dry season or at onset of the wet season, just before crops are planted. Where the first burn is incomplete, typically because the vegetation is too moist or weather conditions unsuitable, further burns, usually of piled material, may take place to fully clear the plot (e.g., [53]). In drier regions, where agricultural burning poses wildfire risk, burning may take place slightly earlier in the dry season to mitigate wildfire risk [30].

The limited available data (n = 9) for seasonality of pre-harvest weed or crop burning (A3) mostly relate to examples of sugarcane burning, or weeding of perennial crops (e.g., fruit orchards) in the dry season in precipitation-driven regions. We might also expect some burning of weeds to take place during wetter times of year while crops are growing, though the data do not evidence this. Examples of post-harvest crop residue or weed burning (A4) range greatly in their seasonal timing in precipitation-, combination-, and temperature-driven climate seasonality regions (Figs 5–7). This variation may be explained by the fact that in some places, multiple crops are grown in the same fields annually, e.g., summer and winter wheat crops in the southwest of Russia, and fire is used to prepare land for both [54]. In regions with precipitation- or combination-driven climate seasonality, however, examples of post-harvest crop residue or weed burning are more concentrated at drier times of year, which is commonly when fields are being prepared for planting. For instance, in regions with precipitation-driven climate seasonality, only 10% − 17.5% of fire use practices with this purpose recorded burning taking place in months at the height of the wet season (−6 to −3 in the seasonal cycle), compared with 22.5% − 47.5% in other months of the year.

In regions with precipitation-driven climate seasonality, all other fire use purposes for which we could analyse data, except for burning to facilitate honey gathering (G3), are more likely to take place in the dry season, but some are associated with burning earlier in the dry season than others (Fig 5). Fires associated with wildfire risk reduction (HW2 and HW3) are weighted most heavily towards the early dry season (70% and 78% of fire use practices with these purposes noted burning in month −2 respectively), because they usually take place before wildfire risk increases later in the dry season. Fires associated with hunting and fishing (HF1, HF2, HF3, HF4) and gathering (G2), are weighted slightly towards the mid- to late dry season: the HF category overall peaks at month +1 where 71% of fire use practices reported burning,

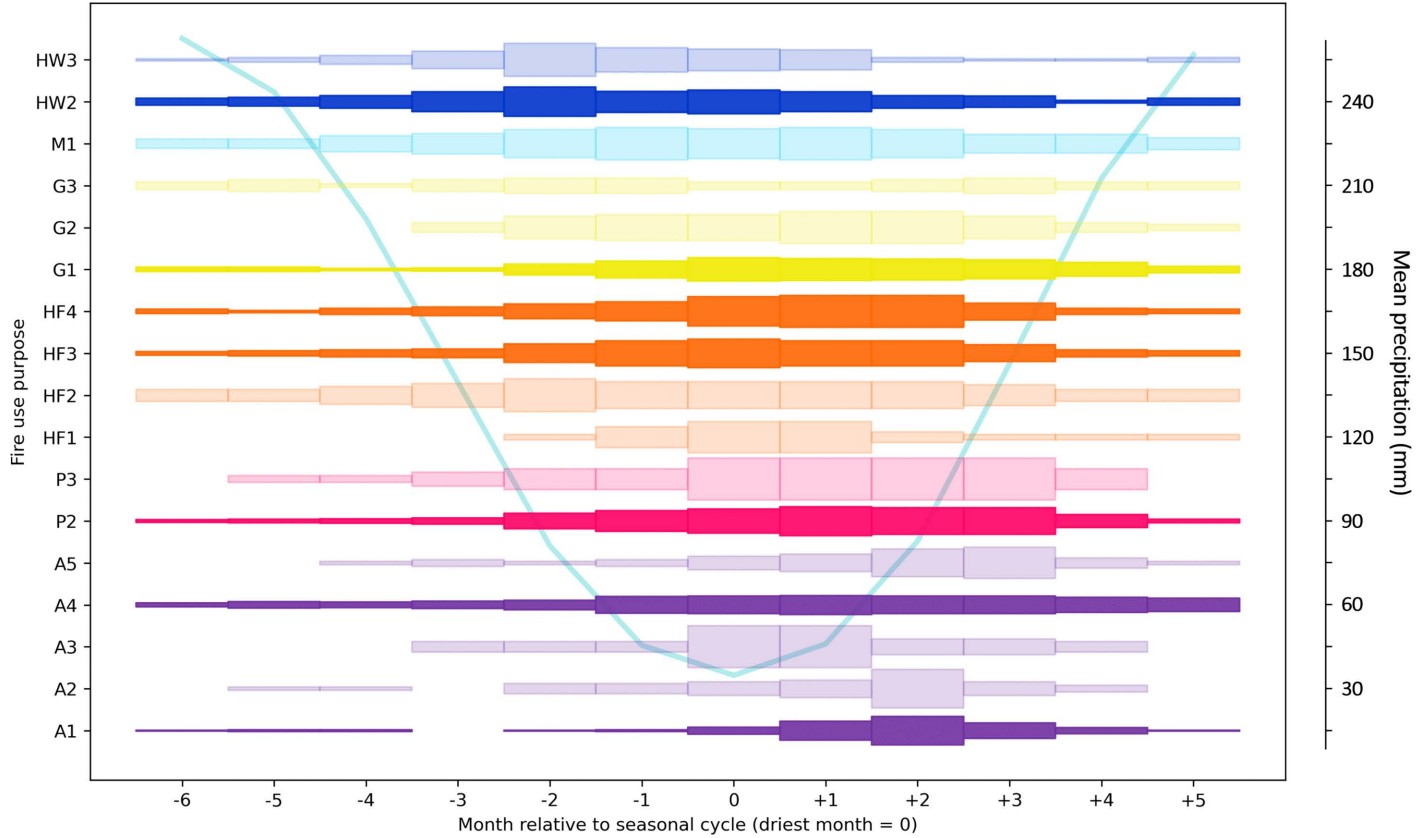

**Fig 5. Seasonality of burning for fire practices recorded in areas with precipitation-driven seasonality.** For each month, the thickness of the shaded bar is proportional to the fraction of records with that fire use purpose that reported burning taking place in that month. See S3 Appendix for data on number of records of burning per month for each fire use purpose. The figure only includes those fire purpose categories for which there were six or more fire use practices with data on the months in which burning takes place. Light shading indicates that at least 6 but fewer than 20 fire use practices contributed data to the analysis for that fire purpose. Dark shading indicates that there were 20 or more fire use practices contributing data. Months are defined relative to the driest month in the annual seasonal cycle (= month 0). The mean precipitation seasonal cycle (mean across case study locations over the time period 1990-2020) is shown in the background.

the period in which vegetation is most likely to enter a state in which it is dry enough to burn. Pasture maintenance (P2) and livestock herding fires (P3), on the other hand, show a weighting towards the late dry season (67% of fire use practices with either of these purposes included burning in month +3). Descriptions in the LIFE database suggest that this is because the coming rains will ensure a flush of fresh growth if burning takes place at this time (see, e.g., [55]).

In regions with combination-driven climate seasonality, though data are more limited, nonetheless for non-agricultural fire uses we also see a weighting towards the driest part of the year, when temperatures are rising (Fig 6). Again, fires associated with wildfire risk reduction (HW3 and HW2) are weighted earlier in the dry period, relative to fires associated with hunting and fishing (HF2 and HF3), improving resource quality for gathering (G1), and maintaining pastures (P2).

For regions with temperature-driven climate seasonality, besides crop residue burning, we only analysed data for months of burning for pasture maintenance fires (P2), which commonly take place in the spring, and sometimes in the autumn (Fig 7). Sources in the LIFE database that described broad seasons but did not specify specific months in which burning occurs, suggest that in northerly latitudes, Autumn and Spring are also common seasons for fires associated with hunting, fishing and gathering (e.g., [42,56,57]).

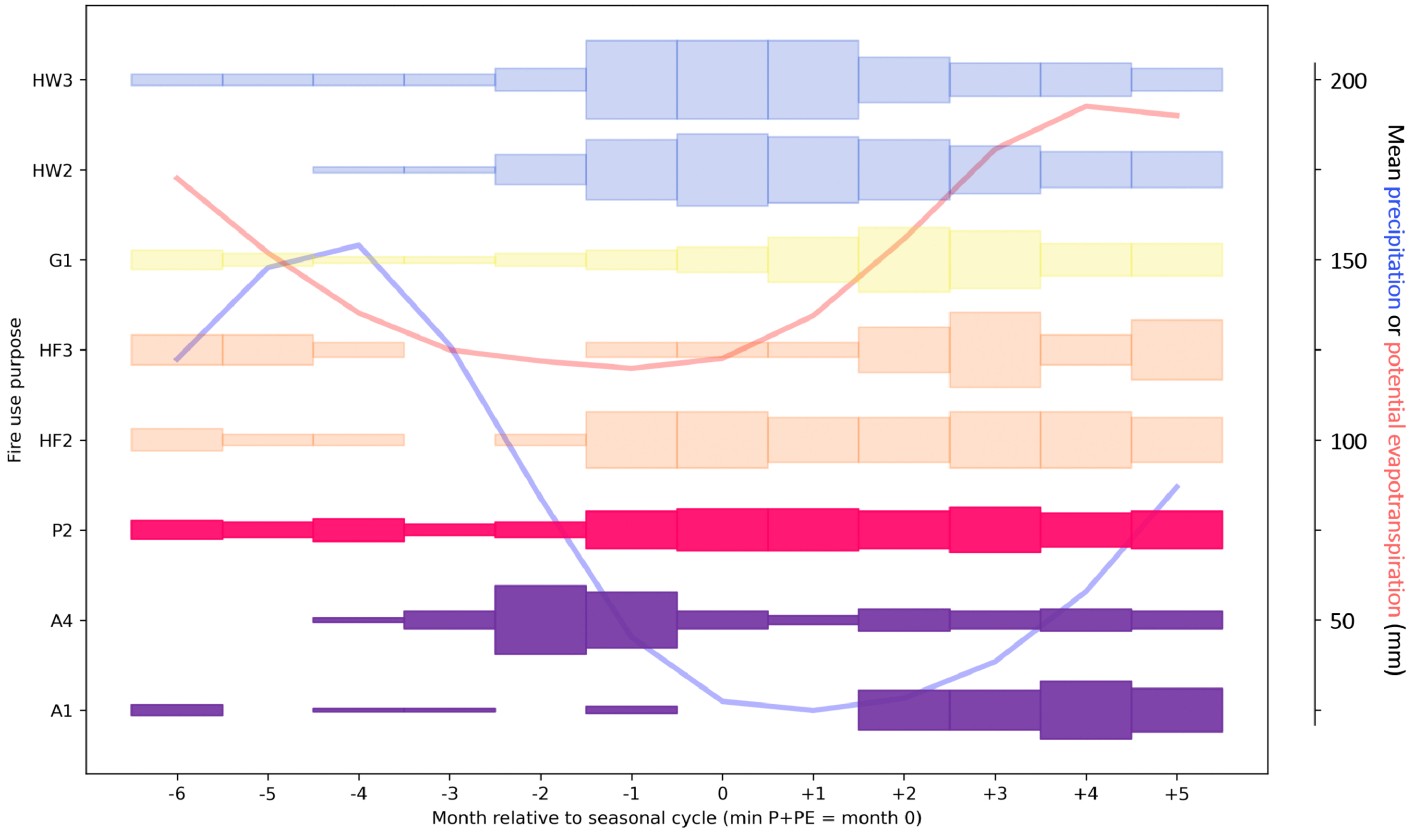

**Fig 6. Seasonality of burning for fire practices recorded in areas with combination-driven seasonality.** For each month, the thickness of the shaded bar is proportional to the fraction of records with that fire use purpose that reported burning taking place in that month. See S3 Appendix for data on number of records of burning per month for each fire use purpose. The figure only includes those fire purpose categories for which there were six or more fire use practices with data on the months in which burning takes place. Light shading indicates that at least 6 but fewer than 20 fire use practices contributed data to the analysis for that fire purpose. Dark shading indicates that there were 20 or more fire use practices contributing data. Months are defined relative to the month with the lowest sum of precipitation and PET in the annual seasonal cycle (= month 0). The mean precipitation and PET seasonal cycles (mean across case study locations over the time period 1990-2020) are shown in the background.

For some of our other fire use purpose categories, we can make some assumptions about their seasonality. We can assume that backing fires set to meet and suppress wildfires (HW4) are most common at times of year with the highest wildfire risk. In regions with precipitation-driven climate seasonality, for instance, this would be mid- to late dry season. Meanwhile, fires associated with rituals and ceremonies (S3) may well show strong seasonality in different locations, if tied to specific annual events, but the seasonal timing of such events is likely to vary widely in different cultural settings.

## 4  Discussion

Our results identify relationships between the underlying purpose of small-scale livelihood and cultural fire use and the resulting fire return interval, burned area, and seasonal timing. Agricultural fires, for example, show strong seasonality linked to planting cycles, and are consistently small (<2 ha) compared to other types of anthropogenic fires, but vary greatly in return interval. However, data in LIFE suggest that fires set for the same purpose can also range greatly in burned area (e.g., pasture maintenance fires) and fire return interval (e.g., swidden clearance fires)

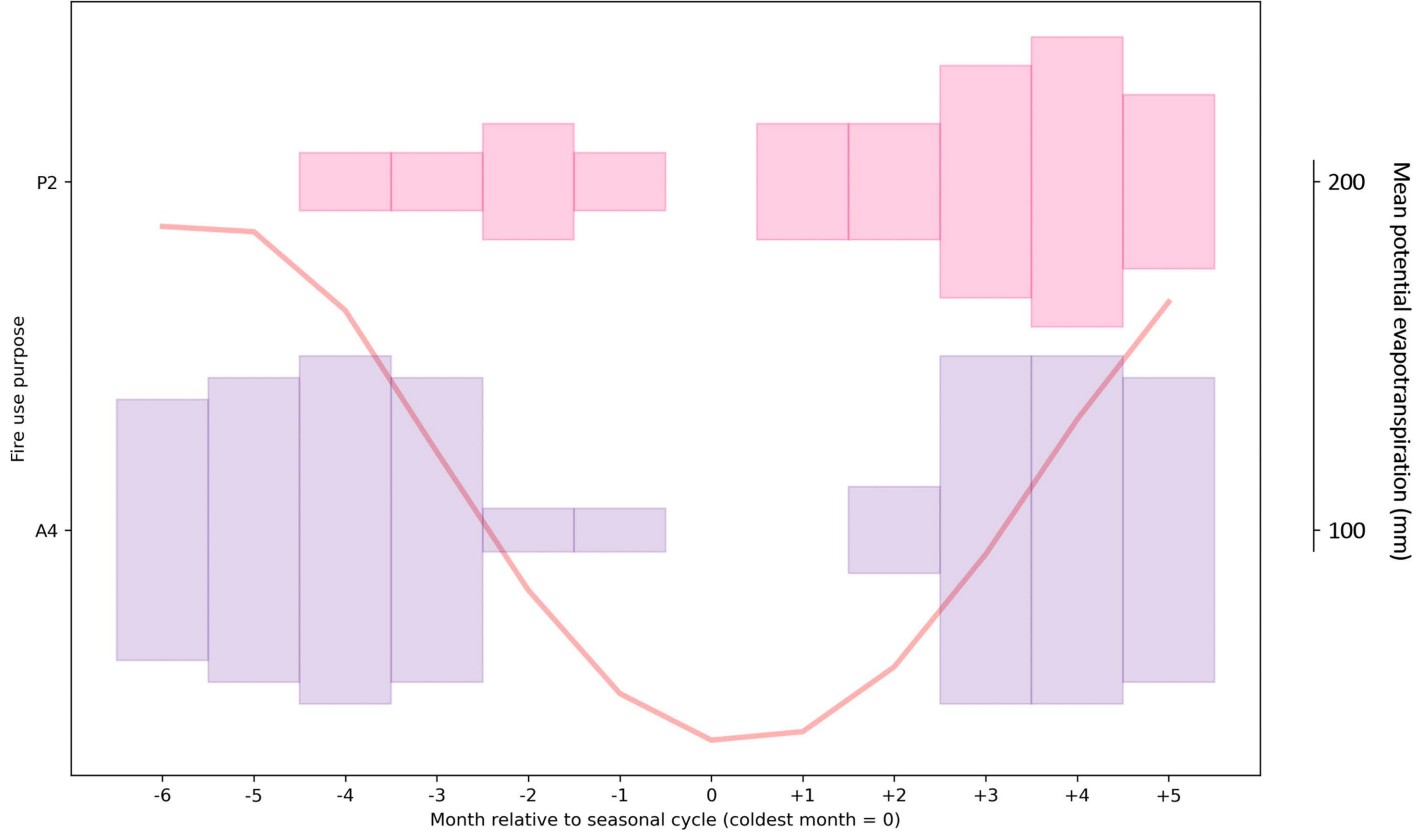

**Fig 7. Seasonality of burning for fire practices recorded in areas with temperature-driven seasonality.** For each month, the thickness of the shaded bar is proportional to the fraction of records with that fire use purpose that reported burning taking place in that month. See S3 Appendix for data on number of records of burning per month for each fire use purpose. The figure only includes those fire purpose categories for which there were six or more fire use practices with data on the months in which burning takes place. Light shading indicates that at least 6 but fewer than 20 fire use practices contributed data to the analysis for that fire purpose. Dark shading indicates that there were 20 or more fire use practices contributing data. Months are defined relative to the month with the lowest PET in the annual seasonal cycle (= month 0). The mean PET seasonal cycle (mean across case study locations over the time period 1990-2020) is shown in the background.

(Table 2, Figs 3 and 4). Some spatiotemporal variation may exist because fire use sometimes fulfils multiple purposes simultaneously. For example, hunting and gathering fires may also be associated with reducing fuel loads at the landscape scale to reduce wildfire risk reduction. Yet, much of this variation exists because fire-use purposes are only one factor shaping anthropogenic fire. Fire use varies in space and time, even on decadal timescales. Many case studies of swidden agriculture in the LIFE database, for instance, suggest that fallow periods (and thus fire return intervals) have declined since the early 1990s, while the seasonality of swidden fires is being affected by climate change [8].

Understanding such complexities in the quantitative signatures of anthropogenic fire will be important to interpret the improved detection of small fires in remote-sensing derived global burned area products. For example, the GloCAB cropland fire product of Hall et al. [58], which contributed to GFED5, suggests that in recent decades cropland burned area has declined globally at a similar rate to overall vegetation fire [16]. Yet the data in LIFE point to a more complex picture for the case of swidden fires: whilst the spatial extent of swidden fire is declining [8], reduced fallow periods suggest where it persists it is leading to an increase in burned area. Indeed, the GloCAB cropland burned area product draws on ground-truth data in regions where cropland burning is driven by disposal of residues in permanent agriculture [59], rather

than shifting cultivation. Our results point to contrasting seasonality between the two fire use purposes, with swidden fires (purpose A1) occurring in a much narrower interval of the driest months than crop residue burning (purpose A4; Fig 5). Such complexities may confound the current satellite-based detection approach and point to areas where such products may be improved. For example, future iterations of cropland fire products could use LIFE data and seasonality patterns as a means of evaluating and sense-checking fire detection.

The method for capturing the seasonality of human fire use presented here also has applications for current global models of vegetation fire. For instance, models in the first Fire Model Intercomparison Project (FireMIP) uniformly placed the peak of burning in the North Hemisphere tropics around two months later than suggested by satellite-derived observations [20]. Human fire use for fuel load reduction (HW2) is common in much of this region (e.g., [60]), and our results suggest this fire use is associated with relatively large burnt areas (Table 2, Fig 4), quite frequent return intervals (Table 2, Fig 3), and often precedes the peak of the dry season, with the most commonly reported timing being two months before the driest month (month −2) in precipitation-driven climates (Fig 5). Incorporating a representation of certain categories of intentional human fire use, in which the occurrence and seasonal distribution is parameterized relative to the precipitation seasonal cycle following the distributions shown here, may therefore help to correct this consistent model seasonality bias in the tropics. Perhaps more fundamentally, no FireMIP model could reproduce the global distribution of fire sizes better than a random null model [20]. Given human fire use often results in small fire sizes, it is likely this bias will not be fixed without incorporating a better representation of anthropogenic burning.

As such, data presented here offer insights into model structures that move beyond simple population density & GDP approaches [19,61]. For example, quantitative data from case studies could be used to develop some general rules for global fire models, distinguishing the properties of anthropogenic fires by their intended purpose, as were previously developed from qualitative information for the LPJ-LMfire model of pre-industrial fire [62], and is in early-stage development for the present-day [60,63]. Alternatively, a first step for improving representation of human fire in DGVMs could be to incorporate our projections of seasonality into the existing metrics of human activity. Overall, results presented here represent an initial quantitative basis to support further local and landscape-scale modelling of human fire interactions across diverse socio-ecological contexts.

However, our analysis considers only small-scale livelihood and/or cultural fire use. Given the small size of many of these fires they will likely be a major contributor to previously-undetected burned area, but fire use by larger landowners and state agencies should also be considered in understanding the contributions of anthropogenic fire to global fire regimes. Fire use data for such land users is more widely available in government and Non-Governmental Organizations' plans and reports than scientific literature [23]. Understanding contemporary and future fire regimes must also consider human fire suppression and mitigation, including both intentional activities such as creating firebreaks or firefighting, and the indirect effects of land use change. Global synthesis of case study data, including the data in LIFE, can shed light on these practices and enable representation beyond indirect metrics such as road density.

Furthermore, except for agricultural fires (vegetation clearance for swidden especially), our study also highlights the limited quality and quantity of quantitative data available from case studies regarding small-scale livelihood and/or cultural fire use. Burned area data are only available for 10% of the fire use practices in the LIFE database, fire return interval data for 18% and months of burning for 37%, and the limited available data meant we could not look for spatiotemporal patterns for many of our fire use purpose categories. Moreover, of the data available for burned area and fire return interval, many are approximate values, not based on field measurements. The data also show a limited geographical distribution, with certain areas, e.g., Central Africa, Eastern Europe and Russia, lacking data (Fig 1). This paucity of quantitative data may reflect several factors. First, most researchers conducting case study research with fire users are qualitative researchers, whose research questions do not drive them to collect quantitative data on fires. Second, collecting accurate field data on the burned area, return interval and seasonality of human-set fires is difficult, and ideally requires a combination of qualitative research methods with remote-sensing and field verification (see, e.g., [44,64]). Third, in many parts of the world,

regulations ban or restrict fire use, such that many anthropogenic fires are set illegally [65]. This has practical and ethical ramifications for data collection about such fires. Improving the availability of quantitative data on anthropogenic fires globally is, thus, not simple. Researchers must also avoid reproducing historical inequalities in fire management, and consider the research priorities of fire users themselves, which may not align neatly with global data gaps [66–68].

Despite this, we have identified broad patterns in seasonal timing as well as the distribution of fire sizes for some prevalent categories of small-scale livelihood and cultural fire use. Our results highlight variations in fire activity throughout the year dependent on seasonal activities and climate. However, the relationships between fire-use purpose, fire return interval, burned area and seasonality suggest that understanding and modelling small-scale livelihood and cultural fire should go beyond considering fire-use purposes alone. Case studies demonstrate that the locations and timing of fire use are influenced, for instance, by vegetation type, topography, state and local governance, economic pressures, and traditional knowledge. Future qualitative analysis of case studies in the LIFE database will shed light on some of these drivers of human fire use, with applications for modelling and interpretation of the latest fine-scale remote sensing data. But our results also highlight that systematic and comprehensive quantitative data on anthropogenic fires is lacking globally. Whilst recognising the complexity of this challenge, improving the quantity and completeness of quantitative data on human fire use, allied with understanding of fire governance drivers in differing contexts around the world, would enable a more appropriate interpretation of new global fire mapping and modelling dynamics into the future.

## Supporting information

**S1 Appendix. Table of fire use purposes, including detailed description of each purpose and the approximate geographical extent of corresponding fire use practices in the LIFE database.**
(DOCX)

**S2 Appendix. Supplementary figures showing detailed case study locations over Africa and south-east Asia, and precipitation and potential evapotranspiration plots.**
(DOCX)

**S3 Appendix. Spreadsheet of case study numbers within each seasonality type that recorded fire use in each month and for each fire use purpose.**
(XLSX)

## Author contributions

**Conceptualization:** Cathy Smith, Jayalaxshmi Mistry.

**Data curation:** Cathy Smith, Matthew Kasoar.

**Formal analysis:** Cathy Smith, Matthew Kasoar.

**Investigation:** Cathy Smith, Oliver Perkins.

**Methodology:** Cathy Smith, Matthew Kasoar.

**Visualization:** Cathy Smith, Matthew Kasoar, Oliver Perkins.

**Writing – original draft:** Cathy Smith, Matthew Kasoar, James D. A. Millington.

**Writing – review & editing:** Cathy Smith, Matthew Kasoar, Oliver Perkins, James D. A. Millington, Jayalaxshmi Mistry.

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
