## [Decision Letter · Decision Letter 0]

16 Apr 2025

PONE-D-25-13919Small-scale livelihood and cultural fire: global spatiotemporal characteristics, and gaps in dataPLOS ONE

Dear Dr. Kasoar,

Thank you for submitting your manuscript to PLOS ONE. After careful consideration, we feel that it has merit but does not fully meet PLOS ONE’s publication criteria as it currently stands. Therefore, we invite you to submit a revised version of the manuscript that addresses the points raised during the review process.

Based on the reviewers' evaluations, minor improvements are needed to enhance the manuscript; therefore, the decision is for a minor revision.

Please submit your revised manuscript by May 31 2025 11:59PM. If you will need more time than this to complete your revisions, please reply to this message or contact the journal office at plosone@plos.org. Please include the following items when submitting your revised manuscript:

We look forward to receiving your revised manuscript.

Kind regards,

Rodrigo Nogueira Vasconcelos, Ph.D.

Academic Editor

PLOS ONE

“The authors are grateful for funding from the Leverhulme Trust under grant RC-2018-023”

Reviewers' comments:

Reviewer's Responses to Questions

**Comments to the Author**

1. Is the manuscript technically sound, and do the data support the conclusions?

Reviewer #1: Yes

Reviewer #2: Yes

2. Has the statistical analysis been performed appropriately and rigorously? 

Reviewer #1: Yes

Reviewer #2: Yes

3. Have the authors made all data underlying the findings in their manuscript fully available?

Reviewer #1: Yes

Reviewer #2: Yes

4. Is the manuscript presented in an intelligible fashion and written in standard English?

Reviewer #1: Yes

Reviewer #2: Yes

5. Review Comments to the Author

Reviewer #1: The manuscript presents a well-structured and rigorous analysis of small-scale anthropogenic fire use. The methodological approach, combining qualitative and quantitative data from a wide range of sources, is both innovative and comprehensive. The integration of spatiotemporal data into global fire modeling is a significant contribution. The manuscript is coherent and logical, with each section building upon the previous one.

1- Minor revisions in the abstract to emphasize key contributions.

2- A clearer overview of the methodology in the introduction to improve linkages.

3- Expand the conclusion with specific recommendations for future research and policy applications.

Reviewer #2: General Assessment

This manuscript addresses an important and underrepresented topic in global fire science: the spatiotemporal patterns and data gaps related to small-scale livelihood and cultural fire use. It draws on a unique global dataset (the LIFE database) and presents both quantitative and qualitative analyses that are relevant for improving fire models and remote sensing interpretations. The authors have clearly responded to previous reviewer concerns by improving clarity, transparency, and methodological detail throughout the manuscript. While the study remains limited by the availability and quality of existing case study data, it makes a valuable contribution by identifying global patterns and data gaps in anthropogenic fire use. The revised manuscript is substantially improved and, with minor revisions, is suitable for publication.

1. Technical Soundness, Data Support for Conclusions, and Statistical Analysis

The manuscript is technically sound and methodologically well-structured for its exploratory goals. The authors apply clear and justified criteria for selecting data from the LIFE database and transparently describe how central estimates (mean, median, approximate values) were derived. The treatment of seasonality using a climatologically normalized framework is a strength of the study, allowing for comparisons across hemispheres and climate zones.

While no inferential statistical tests are applied, the descriptive statistics are appropriate for the research aims. Thus, while the manuscript does not include hypothesis testing or predictive modeling, it is technically soundness for the purpose it sets out to achieve.

2. Clarity, Structure and English Language Quality

The manuscript is well written, clear, and generally easy to follow.

3. Figures and Visualization

Figure 1 remains difficult to interpret. It attempts to display multiple variables (case study location, data type, fire-use purpose) on a single map using color and background shading. This results in visual clutter and makes interpretation challenging—particularly in regions with overlapping data points and similar color hues.

4. Final Recommendation

Recommendation: Minor Revisions

This is a strong and timely contribution that brings new data and insight into the role of small-scale and cultural fire use in global fire regimes. With minor revisions—primarily to improve figure clarity and streamline some dense text—it will be suitable for publication in PLOS ONE.

6. PLOS authors have the option to publish the peer review history of their article (what does this mean?). If published, this will include your full peer review and any attached files.

Reviewer #1: **Yes:**Aldnira Tolentino Nogueira

Reviewer #2: No

---

## [Author Response · Author response to Decision Letter 1]

17 Oct 2025

We greatly appreciate the time and efforts of both reviewers, and are encouraged by their supportive comments, which we hope we have addressed satisfactorily below.

Reviewer #1:

The manuscript presents a well-structured and rigorous analysis of small-scale anthropogenic fire use. The methodological approach, combining qualitative and quantitative data from a wide range of sources, is both innovative and comprehensive. The integration of spatiotemporal data into global fire modeling is a significant contribution. The manuscript is coherent and logical, with each section building upon the previous one.

1- Minor revisions in the abstract to emphasize key contributions.

2- A clearer overview of the methodology in the introduction to improve linkages.

3- Expand the conclusion with specific recommendations for future research and policy applications.

Authors' response: We are grateful to the reviewer for their very positive appraisal of our work! As suggested, we have made minor text additions to the abstract and the final paragraphs of the introduction to better highlight the applications of this analysis, and introduce the methodological aims. We remain open to any further suggestions of specific improvements that could improve clarity or flow of the text!

Regarding the final recommendation, this prompted some discussion among the authors and ultimately we felt that this manuscript is probably not the right place to try and make specific recommendations of policy applications. We believe that the methodology and analysis of case study data can enable significant improvements to the representation of human fire use in coarse-resolution land surface models, however in isolation these models do not necessarily then provide improved utility for making policy and landscape management decisions. For example, while we might use outputs from such models to identify where interventions might be most beneficial (from a global earth system perspective), nonetheless the context and circumstances of the local situation are also critical to understand the feasibility and impacts on the ground, and there may be additional dimensions (e.g. cultural or historical) which are not well captured in the case study data and quantitative relationships that emerge. So, while there is need and utility in analysing and synthesising local case studies to advance understanding globally, we want to also still be clear on the uncertainties and gaps in our current data, and thus limitation of our models (as we hope this paper does).

Reviewer #2:

General Assessment

This manuscript addresses an important and underrepresented topic in global fire science: the spatiotemporal patterns and data gaps related to small-scale livelihood and cultural fire use. It draws on a unique global dataset (the LIFE database) and presents both quantitative and qualitative analyses that are relevant for improving fire models and remote sensing interpretations. The authors have clearly responded to previous reviewer concerns by improving clarity, transparency, and methodological detail throughout the manuscript. While the study remains limited by the availability and quality of existing case study data, it makes a valuable contribution by identifying global patterns and data gaps in anthropogenic fire use. The revised manuscript is substantially improved and, with minor revisions, is suitable for publication.

1. Technical Soundness, Data Support for Conclusions, and Statistical Analysis

The manuscript is technically sound and methodologically well-structured for its exploratory goals. The authors apply clear and justified criteria for selecting data from the LIFE database and transparently describe how central estimates (mean, median, approximate values) were derived. The treatment of seasonality using a climatologically normalized framework is a strength of the study, allowing for comparisons across hemispheres and climate zones.

While no inferential statistical tests are applied, the descriptive statistics are appropriate for the research aims. Thus, while the manuscript does not include hypothesis testing or predictive modeling, it is technically soundness for the purpose it sets out to achieve.

2. Clarity, Structure and English Language Quality

The manuscript is well written, clear, and generally easy to follow.

3. Figures and Visualization

Figure 1 remains difficult to interpret. It attempts to display multiple variables (case study location, data type, fire-use purpose) on a single map using color and background shading. This results in visual clutter and makes interpretation challenging—particularly in regions with overlapping data points and similar color hues.

4. Final Recommendation

Recommendation: Minor Revisions

This is a strong and timely contribution that brings new data and insight into the role of small-scale and cultural fire use in global fire regimes. With minor revisions—primarily to improve figure clarity and streamline some dense text—it will be suitable for publication in PLOS ONE.

Authors' response: We greatly appreciate the reviewer’s supportive comments on our manuscript. The author group spent some time debating whether there is a way to improve the clarity of Figure 1. For the figure in the main text we could not come up with a better way to present the information globally – the main purpose of the figure is ultimately to show the large-scale distribution and representativeness of the case studies in a concise way. With the existing figure we did already attempt to use a colour palette that makes the symbols distinct from the background shading, and used 3 separate maps for each of the different quantitative properties that we subsequently analyse to avoid the different symbols obscuring each other. Further splitting by the fire-use purpose would result in 24 maps which we feel would probably only increase the clutter. Instead, we have now added two additional supplementary figures (figures S1 and S2 in the revised Appendix B) which zoom in on Africa and south-east Asia - the two regions where the density of case studies is particularly high, which we hope will therefore allow interested readers to see the finer detail and distinguish the individual points over these regions, while still preserving the main Figure 1 as a global overview. Hopefully this addresses the main concern, but we remain open to any alternative suggestions that would improve clarity of either the figure or text!

Additional changes to the manuscript:

Equations (1) and (3) describing the criteria for temperature-driven and precipitation-driven seasonality zones, had erroneously been labelled the wrong way around, i.e. the equation for precipitation-driven seasonality was incorrectly labelled as ‘temperature-driven’, and vice versa. This mistake in the description of the equations has now been corrected, and we apologize for the error. Only the wording in the description of the equations was incorrect; the equations were correctly applied in the analysis and none of the results are affected.

---

## [Editor Report · Decision Letter 1]

9 Dec 2025

Small-scale livelihood and cultural fire: global spatiotemporal characteristics, and gaps in data

PONE-D-25-13919R1

Dear Dr. Kasoar,

We’re pleased to inform you that your manuscript has been judged scientifically suitable for publication and will be formally accepted for publication once it meets all outstanding technical requirements.

Kind regards,

Kristofer Lasko, PhD

Academic Editor

PLOS One

Additional Editor Comments (optional): Ensure you either delete your empty "Acknowledgments" section, or populate it as needed.
---

## [Editor Report · Acceptance letter]

PONE-D-25-13919R1

PLOS One

Dear Dr. Kasoar,

I'm pleased to inform you that your manuscript has been deemed suitable for publication in PLOS One. Congratulations! Your manuscript is now being handed over to our production team.

Kind regards,

on behalf of

Dr. Kristofer Lasko

Academic Editor

PLOS One